# Comparative Proteomic Analysis of Drug Trichosanthin Addition to BeWo Cell Line

**DOI:** 10.3390/molecules27051603

**Published:** 2022-02-28

**Authors:** Yajun Hu, Jun Yao, Zening Wang, Hui Liang, Cunyu Li, Xinwen Zhou, Fengying Yang, Yang Zhang, Hong Jin

**Affiliations:** 1Shanghai Stomatological Hospital & School of Stomatology, Fudan University, 356 Beijing East Road, Shanghai 200001, China; yjhu17@fudan.edu.cn; 2Department of Chemistry, Fudan University, 220 Handan Road, Shanghai 200433, China; yaojun123@fudan.edu.cn; 3Institutes of Biomedical Sciences, Fudan University, 138 Yixueyuan Road, Shanghai 200032, China; 21211510002@fudan.edu.cn (Z.W.); leecyts@163.com (C.L.); zhouxinwen@fudan.edu.cn (X.Z.); yyan2009@126.com (F.Y.); 4College of Pharmacy, Guangxi Medical University, 22 Shuangyong Road, Nanning 530031, China; lhhh02@163.com

**Keywords:** TCS, MTT, MALDI-TOF/TOF MS, proteomic, BeWo cell line

## Abstract

Trichosanthin (TCS) is a traditional Chinese herbal medicine used to treat some gynecological diseases. Its effective component has diverse biological functions, including antineoplastic activity. The human trophoblast cell line BeWo was chosen as an experimental model for in vitro testing of a drug screen for anticancer properties of TCS. The MTT method was used in this study to get a primary screen result. The result showed that 100 mM had the best IC_50_ value. Proteomics analysis was then performed for further investigation of the drug effect of TCS on the BeWo cell line. In this differential proteomic expression analysis, the total proteins extracted from the BeWo cell line and their protein expression level after the drug treatment were compared by 2DE. Then, 24 unique three-fold differentially expressed proteins (DEPs) were successfully identified by MALDI-TOF/TOF MS. Label-free proteomics was run as a complemental method for the same experimental procedure. There are two proteins that were identified in both the 2DE and label-free methods. Among those identified proteins, bioinformatics analysis showed the importance of pathway and signal transduction and gives us the potential possibility for the disease treatment hypothesis.

## 1. Introduction

TCS is an abortifacient protein purified from Tian-hua-fen, a traditional Chinese medicine obtained from the root tubers of Chinese trichosanthes. It is a ribosome-inactivating protein with high antitumor activity [1] and endowed with multiple vital biological activities, including anti-HIV and antitumor processes [2]. Furthermore, TCS has been found to inhibit a variety of tumors, including cervical cancer, choriocarcinoma, leukemia/lymphoma, gastric cancer, colon cancer, hepatoma, breast cancer, and prostate cancer. TCS is a prescription drug in China for gynecological use in various diseases such as ectopic pregnancies, hydatidiform moles, chorionic epithelioma, and abortions [3]. It is normally used as a pharmaceutical material for midterm abortion. It possesses high antitumor activity, through a ribosome-inactivating mechanism, and apoptosis induction to present great promise for cancer therapy [4]. TCS displays a broad spectrum of biological and pharmacological activities including antitumor, antivirus, and immune-regulatory activities. Previous studies have indicated that TCS exerts antivirus, immuno-regulation and broad-spectrum antitumor pharmacological activities [5]. It has been previously reported that TCS in combination with paclitaxel can efficiently reverse MDR, revealing its potential for antiMDR cancer therapy [6]. However, the antitumor application of TCS is restrained by its short half-life owing to rapid renal clearance related to its relatively small size (27 kDa) as well as the poor intracellular delivery efficiency [3,7]. The anticancer activities of TCS are associated with the induction of specific changes of the cytoskeleton configuration by reducing the expression of tubulins [8]. 

TCS has shown an ability to eliminate human choriocarcinoma cells in vitro [9,10]. The toxic mechanisms of TCS on tumor cells also include inhibition of the proliferation and induction of apoptosis of tumor cells, and the detailed mechanism varies in different tumor cells [9]. Further research on the antitumor activities of TCS may not only shed light on cancer therapy but also on new pharmacological properties of ancient Chinese medicines. 

Choriocarcinoma (Gynecology), or chorionic carcinoma, is a kind of high-grade malignant tumor, most of which are related to pregnancy, which can be secondary to hydatidiform moles, abortion, ectopic pregnancy, and term delivery. It is one of the most common types of gestational trophoblastic tumor. Most cases are secondary to normal or abnormal pregnancy, clinically also known as gestational choriocarcinoma, which is highly malignant. 

The human trophoblastic BeWo cell line is the cell line with the closest biological properties to the originally separated trophoblast tumor cell line. BeWo cells can also be killed according to the result of the MTT method. It has been reported that TCS is used in several human disease treatments without any toxic effects. It has been reported that a molecular complex is formed between carcinogens and TCS, in both in vitro and in vivo studies, suggesting that this complex formation might be due to the antigenotoxic effect of TCS [11,12]. The antigenotoxic effect of TCS might be related to inhibition of P450 enzymes by involving the bioactivation of carcinogens to the molecular complex formation between carcinogens and TCS [9]. In addition, TCS can impact on placental syncytial trophoblasts. It can also deactivate the intracellular polysaccharides, leading to cell death, degeneration, and necrosis [7].

Powerful proteomic techniques with high-resolution, high-throughput, real-time protein expression analysis can provide novel ideas for detecting important proteins during the pathological process. In order to reveal the potential mechanisms of the antitumor effect of TCS and to screen out more important proteins related to the addition of TCS to the BeWo cell line, two-dimensional electrophoresis (2DE) and MALDI-TOF/TOF MS were first used to find some distinct differentially expressed proteins before and after the above drug addition to the BeWo cell line. The information of all the identified differentially expressed proteins can be obtained through a search of the human database. With bioinformatic analysis, their molecular function and the pathways these proteins participate in can be retrieved to help understand their roles in cancer. The results could provide guidance for the study of the mechanisms of or treatments for trophoblast tumors.

Here, we chose this natural product to screen its effect on protein level to help treat trophoblast tumors. To obtain a proteomic insight into the antitumor effect and mechanism of TCS on BeWo cells, both 2DE and label-free method were used. The traditional 2DE technique was introduced more than three decades ago [13,14,15,16] and is now still a very widely used method for global proteomic separation and visual protein-expression exhibition [17,18,19,20]. Proteins are separated by two different physicochemical principles, isoelectric point and molecular weight. Following in-gel digestions, the digested peptides were identified by MALDI-TOF/TOF MS. As with label-free method [21], samples were analyzed sequentially by LC-MS/MS. The high-resolution MS1 ion peaks were extracted as input for calculating the quantitative features while the MS2 scans were often utilized merely to assign peptide IDs. Despite being used for more than a decade, there have been booming breakthroughs in label-free method innovation [22,23]. We used the traditional label-free method to identify differentially expressed proteins quantitatively. Comparing the results from 2DE and label-free methods, about 1/3 proteins appeared in both methods. To identify the disease-associated proteins, we screened out the DEPs related to choriocarcinoma or general cancer and attempted to study the mechanisms involved. 

## 2. Materials and Methods

### 2.1. Cell Culture 

BeWo cells were cultured according to the protocol recommended by the American Type Culture Collection (ATCC) [24]. When cells were grown to 50% confluence, fresh medium containing the vehicle (dimethyl sulfoxide, DMSO) was added and cells were incubated in a humidified atmosphere of 95% air and 5% CO_2_. All experiments were performed at least three times independently and in triplicate. The conditioned media and cells were collected and stored at −80 °C until use.

### 2.2. MTT Method 

The MTT assay was performed as described by Mosmann [25] with the modifications suggested by Denizot and Lang [26]. A filtered MTT stock solution (5 mg/mL, distilled water) was sterilized and kept for no more than 2 weeks at 4 °C or a new stock solution would be prepared. To start the reaction, stock solution was added to growing cultures to a final MTT concentration of 0.5 mg/mL. The mixture was incubated for 16 h on a shaker (160 rpm at 20 °C). Cells were pelleted by centrifugation in Eppendorf tubes (15,000× *g*, 5 min) and the medium was removed. Then, 500 µL of 1-propanol was added to the cells. Lysed cells and debris were pelleted (15,000× *g*, 5 min) and 100 µL of the supernatant was transferred into a 96-well plate. The optical density was measured with a spectrophotometer (M5, Molecular Devices, San Jose, CA, USA) at 600 nm, with 690 nm as a reference read-out. A blank with 1-propanol alone was measured and subtracted from all values. 

### 2.3. Sample Preparation

5 × 10^6^ BeWo cells were harvested, washed with PBS, and centrifuged at 1000× *g* for 10 min. The supernatant was carefully removed. Then, the precipitation was mixed and gently shaken with lysis buffer containing 7 M urea, 2 M thiourea, 4% CHAPS, 100 mM DTT, 0.5 mM PMSF, and 1 mM protease inhibitor cocktail. The suspended cells were lysed using ultrasonic sonicator (Qsonica, Newtown, CT, USA) on the ice and the 3-min-long lysis program was disrupting for 5 s at 50% amplitude followed by 10 s interval. Then, the lysis buffer was put on the ice for 40 min and cellular debris was removed by centrifugation at 15,000× *g* at 4 °C for 45 min. The resulting samples were treated immediately or stored at −80 °C until analysis. Protein concentration in the sample was measured by the modified Bradford method [27]. 

### 2.4. Two-Dimensional Gel Electrophoresis (2DE)

The first dimension, isoelectric focusing (IEF) electrophoresis, was performed using 18 cm, pH 3–10 non-linear IPG strips (Bio-Rad, Hercules, CA, USA). Each sample was diluted in rehydration buffer, which contained 7 M urea, 2 M thiourea, 4% CHAPS, 65 mM DTT, 0.5% ampholytes at pH 3–10 and Bromophenol blue to a final volume of 340 mL containing 250 μg protein. The IPG strip was rehydrated for 14 h at room temperature. Isoelectric focusing with Protean IEF Cell (Bio-Rad, Hercules, CA, USA) was performed at 20 °C using the following protocol: 1000 V for 2 h, 4000 V for 1 h, 8000 V for 1.5 h, and keep 8000 V for 9 h. After IEF, the IPG strip was equilibrated in two equilibration solutions for 15 min each with gentle shaking. The first equilibration solution contained 6 M urea, 2% SDS, 20% glycerol, 0.05 M Tris-HCl (pH 8.8), and 2% DTT. In the second equilibration solution, DTT was replaced by 2.5% iodoacetamide. For the second dimension, SDS-PAGE, the IPG strip was transferred to a homogeneous polyacrylamide gel (12%, 200 × 230 × 1.0 mm) and the electrophoresis was performed using a gel running system (Bio-Rad, Hercules, CA, USA) at a current of 40 mA per gel for 4 h (see Appendix A).

### 2.5. Silver Staining and Image Analysis

Proteins were stained with silver nitrate as previously described by Blum et al. [28] with some modifications. The gels were fixed in 45% ethanol and 5% acetic acid solution overnight. Then, the gels were pre-treated in 0.02% Na_2_ S_2_O_3_ solution for 2 min, and subsequently immersed in 0.1% AgNO_3_ solution for 20 min at 4 °C. The signal development was performed with developing solution (2% Na_2_CO_3_ and 0.04% formaldehyde) and finally terminated with a stop solution (5% acetic acid). Stained 2D gels were visualized using LabScan scanner (GE Healthcare, Uppsala, Sweden) and analyzed with Image Master Platinum 6.0 software (GE Healthcare, Uppsala, Sweden) on the optical property of silver-stained spots. The protein spots that showed statistical difference (*p* < 0.05) between cells before and after drug addition were selected for further identification.

### 2.6. In-Gel Digestion and Mass Spectrometric Identification

Selected spots were excised from gels using Ettan Spot Picker (GE Healthcare, Uppsala, Sweden). The de-staining procedure was carried out by washing the spots with a solution of 15 mM potassium ferricyanide and 50 mM sodium thiosulfate (1:1) for 20 min at room temperature. While the spots turned almost colorless, the gel pieces were dehydrated with ACN for 15 min. Then, 50 µL 10 mM DTT solution was added and the solution was incubated at 56 °C for 30 min. Then, 50 mM IAA solution was used to remove excess DTT in darkness for 30 minutes. Then, the ACN-dried gel pieces were swollen in a digestion buffer containing 20 mM NH_4_HCO_3_ and 12.5 ng/mL trypsin (Promega, Fitchburg, WI, USA) at 4 °C. After incubation for 30 min, the samples were incubated at 37 °C overnight. The released peptides were eluted with 50% ACN/0.1% TFA and the peptide solution was dried with N2. For MALDI-TOF/TOF MS analysis, 0.8 µL matrix (5 mg/mLα-cyano-4-hydroxy-cinnamic acid diluted in 50% ACN/0.1% TFA; MilliporeSigma, St. Louis, MO, USA) was added and mixed at least 30 times to dissolve the peptides.

MALDI MS measurements were carried out on a 5800 proteomic analyzer (Sciex, Framingham, MA, USA). Mass spectrum was obtained on a mass range of 800–3200 Dalton by using a laser beam (337 nm, 200 Hz) as ionization source. The instrument was used in reflector positive mode with an acceleration voltage of 20 kV. Trypsin-digested peptides of horse myoglobin were used as external mass standard to calibrate the instrument, and then default calibration was applied on the sample peptides. The TOF–TOF mass spectra were acquired by the Data Dependent Acquisition (DDA) method with 6 strongest precursor ions selected automatically from one MS scan for MS/MS analysis. Protein identification using combined raw data (PMF+ MS-MS) was performed with GPS software (Applied Biosystems; containing MASCOT search engine) against the SwissProt human database (2021-June, 20,396 entries). The mass tolerance was set as 0.3 Da, and MSMS tolerance was 0.4 Da for automatic data analysis. The signal-to-noise ratio (S/N) of fragment ion peaks in tandem mass spectra was 10. False-positive control is listed as follows: (1) 95% confidence level for high-score MS identification and database searching; (2) 2 peptides hit for more specific identification; (3) reverse sequence cut-off for random sequence matching-up.

### 2.7. In-Solution Trypsin Digestion and LC-MS/MS (Label-Free Method)

Acetone precipitation was performed prior to in-solution digestion. Redissolved proteins were reduced with 10 mM DTT for 30 min at 56 °C and alkylated with 55 mM IAA for 30 min at room temperature in darkness. Trypsin-to-protein ratio for digestion was 1:100 at the first 4 h then trypsin was added to a ratio of 1:50 and the reaction was kept overnight.

Nano-LC MS/MS experiment was performed on an HPLC system composed by a nanoAcquity Binary Solvent Manager LC pump and a nanoAcquity Sample Manager (all from Waters Corporation, Milford, MA, USA) connected to an LTQ-Orbitrap XL mass spectrometer (Thermo Fisher Scientific Inc., San Jose, CA, USA). An amount of 10 μg protein digest was used for each sample. Sample was loaded onto an Acclaim PepMap precolumn (0.1 × 20 mm, Thermo Fisher Scientific Inc., San Jose, CA, USA) for 2 min at a flow rate of 8 μL/min. The sample was subsequently separated by an Acclaim PepMapcolumn (0.075 × 150 mm, Thermo Fisher Scientific Inc., San Jose, CA, USA) at a flow rate of 300 nL/min. The mobile phases were 0.1% formic acid (phase A and the loading phase) and 99.9% acenitrile with 0.1% formic acid (phase B). A 90-min linear gradient from 3 to 35% phase B was employed. The separated sample was introduced into the mass spectrometer via nano-electrospray source (Thermo Electron Corporation, San Jose, CA, USA). The spray voltage was set at 1.6 kV and the heated capillary at 200 °C. The mass spectrometer was operated in data-dependent mode and each cycle of duty consisted of one full MS survey scan at the mass range 350~1600 Da with resolution power of 60,000 using the Orbitrap section, followed by MS2 experiments for 10 strongest peaks using the LTQ section. Peptides were fragmented in the LTQ section using collision-induced dissociation with helium and the normalized collision energy value set at 35% and previously fragmented peptides were excluded for 60 s. Triple replicates were performed for each sample.

Raw MS files were processed by MaxQuant version 1.5.2.8 (downloaded from https://www.maxquant.org on 7 June 2019) for database searching and protein quantitation. MS/MS spectra were searched against SwissProt human database (2021-06, 20,396 entries). Precursor “first-search then re-calibration” feature was enabled and first-search tolerance was 25 ppm. Then, mass tolerances of re-calibrated precursors and fragments were 6 ppm and 20 ppm, respectively. Other search parameters included variable modifications of methionine (M) oxidation and N-terminal acetylation at protein N-terminus and fixed modification of carbamidomethylated cysteine (C). Minimal peptide length was set to 7 amino acids and a maximum of two missing cleavages was allowed. For advanced identification features, MS runs were analyzed with the “match between runs” option checked. For this feature, a retention time matching tolerance of 42 s (0.7 min) within an aligning window of 10 min was used. Proteins matching the reverse database (indicated by the header “REV_”) were used to control the false discovery rate (FDR), which was set to 0.01 for both peptide and protein-level identifications. Label-free quantitation method was MaxLFQ [29] with the minimum ratio count set to 2 and minimum number of neighbors to 3. Other parameters included classic normalization type and fast LFQ enabled. The mass spectrometry proteomics data have been deposited to the ProteomeXchange Consortium (http://proteomecentral.proteomexchange.org, last accessed on 7 June 2019) via the iProX [30] partner repository with the dataset identifier PXD030920. 

## 3. Results

### 3.1. MTT Method

Three different concentrations of the drug TCS were screened, and Figure 1 shows the results of the MTT assay as the correlation of IC_50_ and TCS concentration. According to the MTT results, 1000 μg/mL TCS showed the lowest IC_50_ and was chosen for further tests. A picture of 96-well plate is attached as the Appendix A and detailed concentration-dependent MTT result table as the Appendix A.

### 3.2. DE-Derived Proteome

The 2DE graphs before and after drug stimulation are displayed in Appendix A. The gels were scanned and analyzed to compare the overall protein expression patterns of both samples and find the differential points from each other. The iso-electric point and molecular weight distribution of these protein spots were in a wide range. No significantly large area pattern or region difference was found. The BeWo cell line has no metastatic potential, which is the reason it was selected as our experimental cell model to screen the drug. The other reason is that the properties of this kind of cell are close to the original cell culture. 

### 3.3. DEPs Identified by MALDI-TOF/TOF MS

A total of 28 DEPs were successfully identified by MALDI-TOF/TOF MS. All proteins showed 3-fold expression difference or more. The database search gave out 24 unique protein identifications; detailed information for all the identified spots is listed in the Appendix A. These proteins included proliferating cell nuclear antigen (PCNA), peroxiredoxin 1, KRT18 protein, and heat shock 70kDa (Hsp70kDa) protein 5. The peroxiredoxin has an antioxidant function and is related to cell proliferation and signal transduction. It can inhibit the oxidation of other proteins. Peroxiredoxin has the closest relationship with cancer and can eventually become a disease biomarker. This might develop a novel treatment method for carcinoma cancer. Hsp70kDa protein 5 is one of the proteins which are involved in protecting cancer cells against ER stress-induced apoptosis in cultured cells. It protects cancer cells against apoptosis through various mechanisms.

### 3.4. LC-MS/MS-Derived Proteome and Bioinformatics Analysis

LC-MS/MS was used to identify differentially expressed proteins quantitatively. The label-free method is an effective method that supplements 2D gel despite lacking information on intact proteins; both methods are complementary to each other. A series of bioinformatic analyses were performed by using Ingenuity Pathway Analysis (IPA) and various network analysis to deduce key signal pathways concerning the role of TCS in cancer treatment, and this gives us the potential possibility for the hypothesis.

Figure 2 shows the global view of label-free quantification (LFQ) of proteomes and Figure 3 shows the schema of DEPs screening from LC-MS/MS. Figure 2A is the box plot diagram for the LFQ values of all the samples. Figure 2B shows the Orthogonal Partial Least Square Discriminant Analysis (O2PLS-DA) result. In brief, O2PLS-DA is a dimension reduction analysis. The replicates of each sample were clustered close to each other and the discrimination between the samples is indicated by the aggregation of the same or close-type samples and the separation of different-type samples. Figure 2C shows the correlation between the samples. The correlation coefficient between each of the two samples is displayed as a grid element in heat map. The blue rectangle lines mark the border of the correlation matrix between the duplicated samples. All the samples showed considerably high repeatability, while the replicates of sample A prevailed. Figure 2D shows the Hierarchy Clustering. Figure 2E shows a comparison of the protein numbers (Venn graph) among samples A, B, and C. The value on the diagram is the protein number for each sample. There are 506 common proteins identified among all the A, B, and C samples, accounting for about 40% of all the identified proteins. Figure 3A–C shows the volcano plot between each pair of samples, where the *x*-axis is the log2-based fold change, and the *p*-values were calculated by a two-tailed *t*-test. Significantly different proteins must present a fold change of two or larger and a *p*-value less than 0.01. Figure 3D showed the VIP index calculated by the O2PLS-DA model. The protein numbers corresponding to each quantile were calculated. There are 617 proteins whose VIP values are greater than one. Figure 3E,F shows the level of protein expression increased or decreased successively among samples A, B, and C. The DEPs should meet the following criteria: (1) VIP index > 1; (2) significant difference in the volcano plot; (3) among these three samples, A, B, and C, they increased or decreased successively. All DEPs are shown in the Appendix A.

Figure 4 shows the statistics of the Gene Ontology (GO) annotations of DEPs. “Mode” is the protein expression trend, which either increases or decreases successively as described in Figure 3E,F. “Sig” is the significantly different protein shown in Figure 3A–C. As shown in Figure 4B, sub-cellular location, most of the DEPs are located in the mitochondria, but the DEPs between B and A are mainly focused in the ribosome and ER. However, DEPs between C and A are mainly involved in protein folding (Figure 4C). Mode-related DEPs are mainly involved in mRNA processing, metabolism, and transmembrane transport. With the increase of drug dose, the number of down-regulated DEPs was greater than up-regulated DEPs (Figure 4D–F). Down-regulated DEPs between B and A are mainly in the cytoplasm (Figure 4D), while down-regulated DEPs between C and A mainly participate in RNA-binding and nitrogen metabolism (Figure 4E,F). Figure 5 shows the enriched networks that the DEPs participate in. Network Analyst’s pathway-oriented bi-parties network analysis was used (https://www.networkanalyst.ca/, last accessed on 7 June 2019). The hub nodes are KEGG pathway, and the leaf nodes are DEPs. The DEPs mainly take part in spliceosome, ribosome, and tight-junction pathways. However, the DEPs involved in the three pathways are relatively concentrated and rarely participate in other pathways. Spliceosome and ribosome pathways are in the top section of the enriched pathway list, and the DEPs involved in them should be considered as the first rank of candidates. As shown in Figure 5, the pathways with the highest enrichment degrees were spliceosome and ribosome, respectively (Figure 5). They both play important role and are effective during the tumorigenesis. 

RIPs (Ribosome-Inactivating Proteins) are a group of cytotoxin proteins that usually contain an RNA N-glycosidase domain, which irreversibly inactivates ribosomes, thus inhibiting protein synthesis. It is generally believed that its many biological activities act through the inhibition of ribosomes resulting in a decrease in protein synthesis. TCS is a member of the family of ribosome-inactivating proteins and inactivates eukaryotic ribosomes via its N-glycosidase activity. The structural analyses suggest TCS attacks ribosomes by first binding to the C-terminal domain of the ribosomal P protein. As a glycosidase, TCS can inactivate eukaryotic ribosomes by hydrolyzing the N–C glycosidic bond of the adenose at site 4324 in rat 28S rRNA, inhibiting protein synthesis in vitro [30]. This eventually leads to TCS’s action on ribosomes. It has been hypothesized that the rate of entry of TCS into cells to reach ribosomes is an important factor in determining its biological activity.

Splicing is a key link in the transmission of life information from DNA to protein. Spliceosomes, which perform splicing tasks, are multi-component complexes composed of nuclear small RNA and protein molecules with a size of 60 s. This machine can splice pre-mRNA, remove introns, and connect exon sequences into mature mRNA, which is one of the important links of gene expression and regulation. As a dynamic molecular machine, spliceosomes need to be assembled from subunits step by step to complete each splicing event, recognize the splicing sites of RNA precursors, and catalyze the splicing reaction [31]. Spliceosomes are associated with the prognosis of colon cancer related to alternative splicing. The types of proteins are largely related to RNA alternative splicing modification. Only by knowing the spatial location of each RNA, each protein, and even each atom in the spliceosome, can we understand how the splicing process occurs.

Figure 6 shows the IPA analysis results for the DEPs connected to multiple canonical pathways. Figure 6A shows the pathway enrichment of IPA. Cell adhesion and junction-related pathways are highly enriched, for example, Remodeling of Epithelial Adherence Junctions, Epithelial Adherence Junction Signaling, EIF2 Signaling, Germ Cell–Sertoli Cell Junction Signaling, Tight Junction Signaling, and Sertoli Cell–Sertoli Cell Junction Signaling. According to IPA’s Z-score (predicting that the pathway is either activated or inhibited), EIF2 signaling, leukocyte extravasation signaling, and ILK signaling are activated, and the RhoA signaling pathway is inhibited. Figure 6B shows the functional enrichment of DEPs by IPA. RNA post-transcriptional modification is the most significant function, and it is consistent with the results of GO analysis. Figure 6C shows the most abundant three-layer causal network in IPA. Red nodes indicate activation and blue indicate inhibition. DEPs were placed in the middle layer by the software deliberately to demonstrate the relationship in the protein–protein interactions. The upper layer is the upstream proteins regulating DEPs obtained from the IPA database, and the lower layer is the biological results caused by the DEPs. Figure 6D shows the self-built network of DEPs, that is, all the molecules in the network are DEPs. DEPs marked with red stars are hub nodes and proved to be prognostic markers for cancer. This gave us the brief overview of those DEPs from the various directions. Other IPA-enriched networks are presented in the figures.

CTNNB1 (Catenin beta 1) occupied the core position of the enrichment network and IPA network (Figure 6). CTNNB1 is a prognostic marker in colorectal cancer, a cancer-related gene, and a cancer biomarker candidate [32]. It is a key downstream component of the canonical Wnt signaling pathway. In the presence of Wnt ligand, CTNNB1 is not ubiquitinated and accumulates in the nucleus, where it acts as a coactivator for transcription factors of the TCF/LEF family, leading to the activation of Wnt-responsive genes. Through the IPA search, CTNNA1 and CTNNB1 were related to cancer, among which CTNNB1 was a diagnostic marker for many kinds of cancer. If beta catenin is not degraded, it will not stay in the cell for a long time. They will interact with nuclear membrane proteins and enter the nucleus to play an important role. It is not a transcription factor but an important transcription factor activator [32].

## 4. Discussion

The proteomes of BeWo cells differentially stimulated by increasing concentrations of TCS were scanned by 2DE-MS/MS and 2DLC-MS/MS with the label-free method. The samples had good repeatability. At same time, there were also differences between those samples. Using LC-MS/MS, 1253 proteins were detected and relatively quantified in total. A total of 506 proteins (40.38%) were identified in all groups. A total of 101 DEPs were filtered by fold change and *p* value (indicated in Figure 2, detailed data attached as Appendix A) and key DEPs discussed in the following context were listed in Table 1. All the proteins are involved in cancer. PCNA and GAPDH were both identified in the 2DE and label-free methods. Multiple bioinformatic analyses were performed, and the potential pathways and signal transduction networks associated with cancer are also discussed below (diagrams of these pathways and networks are presented in the Appendix A). The results of the enrichment analysis of biological functions and pathways focused on cell death, cancer, glycolysis, and ubiquitination.

PCNA (Proliferating Cell Nuclear Antigen) is a protein that down-regulated after TCS stimulation. It was originally identified as an antigen expressed in the cell nucleus during the DNA synthesis phase of the cell cycle as a co-factor of DNA polymerase delta in eukaryotic cells [33,34]. It mainly distributes in the cyto-plasma and nucleus of cells and participates exclusively in cell signal transduction, which is especially important in the regulation of cancer occurrence. PCNA takes the form of homo-trimer and boosts the productivity of leading strand synthesis during DNA replication. PCNA is post-translationally modified by ubiquitin, and is involved in the DNA mismatch repair pathway (MMR, KEGG ID: ko03420) [35] and also involved in the DNA damage tolerance pathway known as post-replication repair (PRR, ko03030). It is a research hotspot in the mechanism of apoptosis. The expression of PCNA correlated with rectal cancer invasion and lymph node metastasis. The expression level of PCNA proteins in tumor cells reflects the degree of cell proliferation and can be used as an index to evaluate cell apoptosis. It potentially has a proliferative effect and plays a role in cancer development or progression [35]. 

GAPDH (GlycerAldehyde-3-Phosphate Dehydrogenase) is a key enzyme in the glycolysis/gluconeogenesis pathway (map00010). This protein consists of four subunits of 30–40 kDa with a molecular weight of 146 kDa. Regarded as house-keeping gene, it is highly expressed in most cells and tissues. GAPDH has been implicated in many diseases, including those of pathogenic, cardiovascular, degenerative, diabetic, and tumorigenic origins. GAPDH modulates the organization and assembly of the cytoskeleton [36,37]. The protein expression level is generally not affected by recognition sites, phorbol, and other inducers. It has nitrosylase activities and participates in glycolysis and nuclear functions, such as transcription, RNA transport, DNA replication, and apoptosis. GAPDH is a key enzyme in glycolysis that catalyzes the first step of the pathway by converting D-glyceraldehyde 3-phosphate (G3P) into 3-phospho-D-glyceroyl phosphate while glucose consumption is increased in most tumor cells, and glycolysis is therefore up-regulated (Warburg effect). Its nitrosylase activity mediates the cysteine S-nitrosylation of nuclear target proteins such as SIRT1, HDAC2, and PRKDC [38]. Non-glycolytic functions of GAPDH including the regulation of cell death, autophagy, DNA repair, and RNA export were observed in physiological and pathological conditions such as cancer and neurodegenerative disorders. The proposed mechanisms regarding GAPDH-mediated cell death are becoming fundamental for the identification of novel therapies [39]. Understanding the mechanisms can provide insights into how GAPDH can be modulated for therapeutic outcomes [37]. There is also research evidence showing that the GAPDH expression level and tumor metastasis in patients with breast cancer is correlated.

There are 245 proteins with sequentially changed expression levels under increasing concentrations of TCS in these three samples (Figure 3E,F). Among them, NOLC1 and SMHD1 were successively significantly up-regulated, while NP1L1 and SERA were successively significantly down-regulated. NOLC1, SMHD1, and NP1L1 were located in the nucleus, and NOLC1 was involved in transcriptional regulation. All the key proteins discussed in this paper are list in Table 1 and Appendix A.

NOLC1 (nucleolar and coiled-body phosphoprotein 1) was identified and validated as a potential target in multidrug-resistant non-small lung cancer cells. The methylation of this protein was associated with the mechanism of tumorigenesis in hepatocellular carcinoma [40,41].

SMHD1 (structural maintenance of chromosomes flexible hinge domain-containing protein 1) plays a key role in chromosome X inactivation in females by promoting the spreading of heterochromatin [42]. It has ATPase activity and potentially participates in the structural manipulation of chromatin in an ATP-dependent manner in gene expression regulation [43]. It localizes at sites of DNA double-strand breaks in response to DNA damage to promote the repair of DNA double-strand breaks [44,45]. SMHD1 is member of the structural maintenance of chromosomes (SMC), a protein family that plays a key role in epigenetic silencing by regulating chromatin architecture and promotes heterochromatin formation in both autosomes and chromosome X, probably by mediating the merge of chromatin compartments [46].

NP1L1 (nucleosome assembly protein 1-like 1) is a histone H2A/H2B molecular chaperone conserved in various species including yeasts, animals, and plants. It participates in DNA replication and may play a role in modulating chromatin formation and could regulate cell proliferation. By removing/replacing histone H2A/H2B on chromatin, or by regulating nucleosome sliding, NP1L1 can change chromatin structure and regulate chromatin metabolism, thus affecting cell differentiation and ontogeny [47].

SERA (serine-repeat antigen protein) plays an essential role during the asexual blood stage development by controlling the kinetics of merozoite egress from host erythrocytes. It prevents premature rupture of the parasite phorousvacuole and host erythrocyte membranes and plays a role in parasite growth [48].

Screening new anticancer drugs from natural Chinese herbal medicines has become a hot topic in recent years. TCS is a single chain extracted from the root of Trichosanthes kirilowii. TCS could inhibit tumor cell proliferation and angiogenesis. It can induce the process of apoptosis and autophagy, affect the cell cycle, and change the cytoskeleton. Ribosome inactivating protein, belonging to type I ribosome inactivating protein, can attack eukaryotes. The ribosome of the cell inactivates the N-glycosidase activity of its rRNA, resulting in egg production. White matter synthesis was inhibited, which eventually led to the death of eukaryotic cells. At the same time, it is cheap and easily obtained. The recombinant TCS with biological activity has also been successfully developed. It is expected that TCS can be used for the further study of its antitumor effect and mechanisms. As a new antitumor drug, it has been used in clinical therapies and is expected to play a potential role in disease treatment.

## 5. Conclusions

TCS has been proven to be an effective medicine for cancer treatment. However, the mechanism behind this has not been clear until now. Our data highlight the dynamic nature of the comparative proteomic expression level before and after adding TCS to BeWo cell line. Therefore, this comparative proteomic analysis pattern will be a reference tool to study the total protein level changes amid addition to the BeWo cell line and to find potential disease biomarkers. Through multiple proteomic technical routes coupled with bioinformatics analysis, a few important proteins are enriched and focused. Those proteins are also implicated in cancer in the literature.

## Figures and Tables

**Figure 1 molecules-27-01603-f001:**
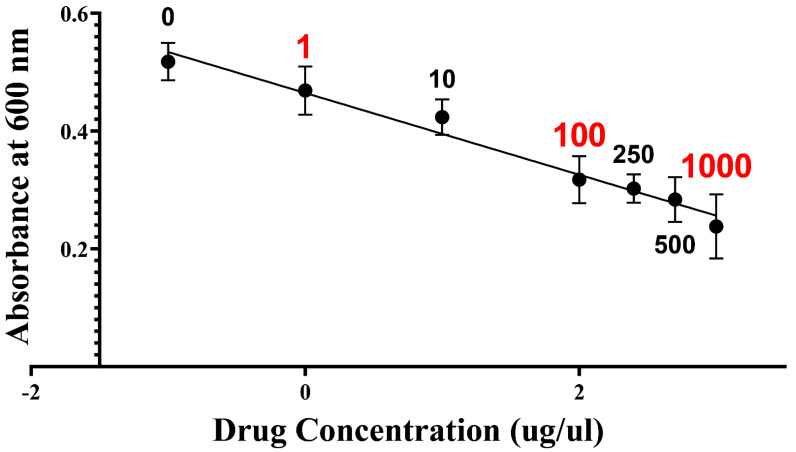
Selection of optimal loading concentration (IC50). Dependency of IC_50_ and TCS concentration used in selecting optimal concentration. The *x* axis was log transformed. Seven different concentrations were tested. Three samples with concentrations labeled red were selected for proteomic analysis. Samples with 1, 10, and 1000 μg/µL added are mentioned in the following context as samples A, B, and C, respectively.

**Figure 2 molecules-27-01603-f002:**
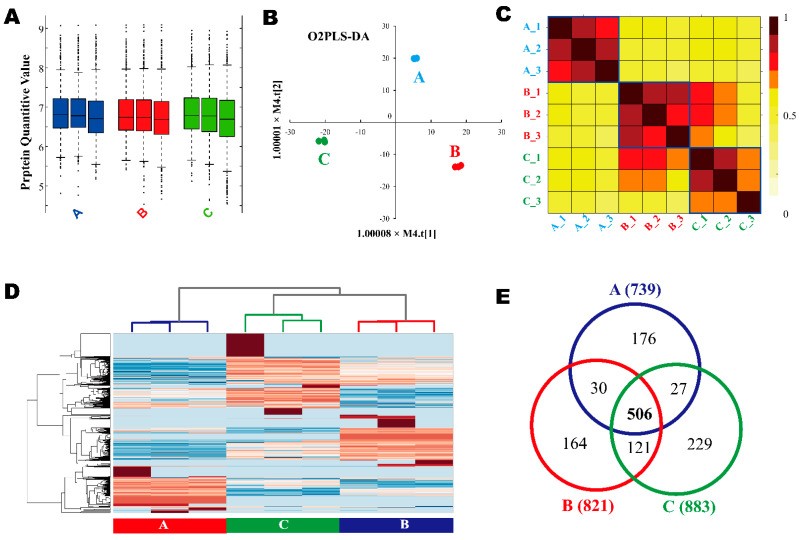
The global analysis of the label-free protein quantitative data. (**A**) Boxplot, (**B**) O2PLS-DA diagram, (**C**) correlation coefficient between samples, (**D**) hierarchy clustering, (**E**) Venn graph. Comparison of protein numbers between samples A, B, and C. There were 506 (40%) common proteins identified among A, B, and C.

**Figure 3 molecules-27-01603-f003:**
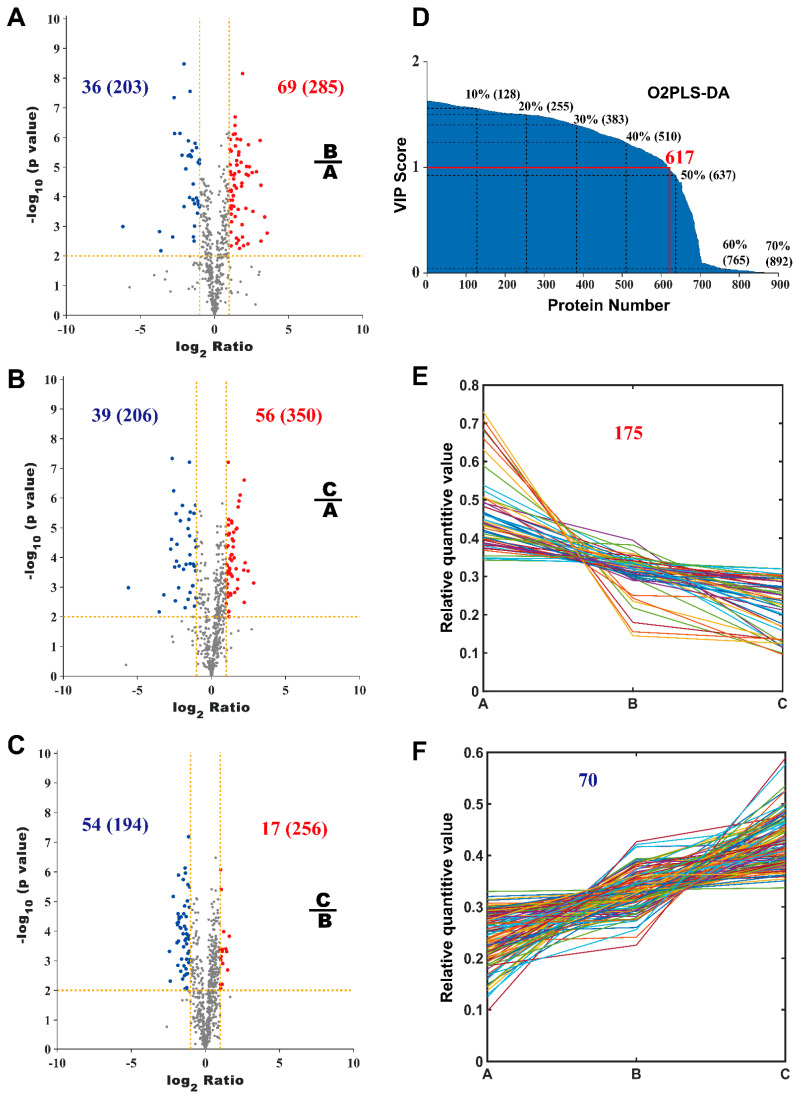
Differentially expressed protein screening. Figures (**A**–**C**) show the volcano plot between every two groups of samples. The *x*-axis is the mean ratio of the protein quantitative values among the samples, and the threshold of the ratio is 2. The *Y*-axis is the *p* value calculated by *t*-test between the samples, and the threshold value is 0.01. The values on the graph are the number of differentially expressed proteins and specifically expressed proteins (in brackets) which were determined. Figure (**E**) shows the increasing level of protein expression for samples A, B, and C successively, and, in contrast, figure (**F**) shows the decreasing level of protein expression. Figure (**D**) shows the VIP index calculated by the O2PLS-DA model. The protein numbers corresponding to each quantile were calculated. There are 617 proteins with VIP > 1.

**Figure 4 molecules-27-01603-f004:**
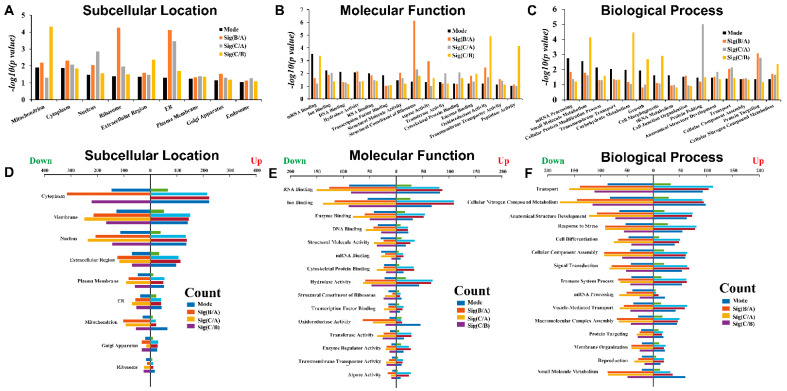
Go analysis of differentially expressed proteins. Figures (**A**–**C**) show the *p* value distributions of subcellular localization, molecular function and biological process of GO. Figures (**D**–**F**) show the protein counts. Mode-related or pair-wisely differentially expressed proteins (B/A, C/A and C/B) are presented as differently colored bars. The differentially expressed proteins are mostly in the mitochondria. Mode-related differentially expressed proteins are mainly involved in mRNA processing, metabolism, and transmembrane transport. The differentially expressed proteins between B and A are mainly focused in the ribosome and ER. In the subcellular localization and biological process of GO, the differentially expressed protein number of mRNA-related processes are the highest. The proteins that significantly up-regulated and down-regulated proteins belong to different GO categories. The significant differentially expressed proteins between B and A are mainly involved in the transport function. In cytoplasm, the numbers of significantly up-regulated proteins between C and A are much higher than those of down-regulated proteins.

**Figure 5 molecules-27-01603-f005:**
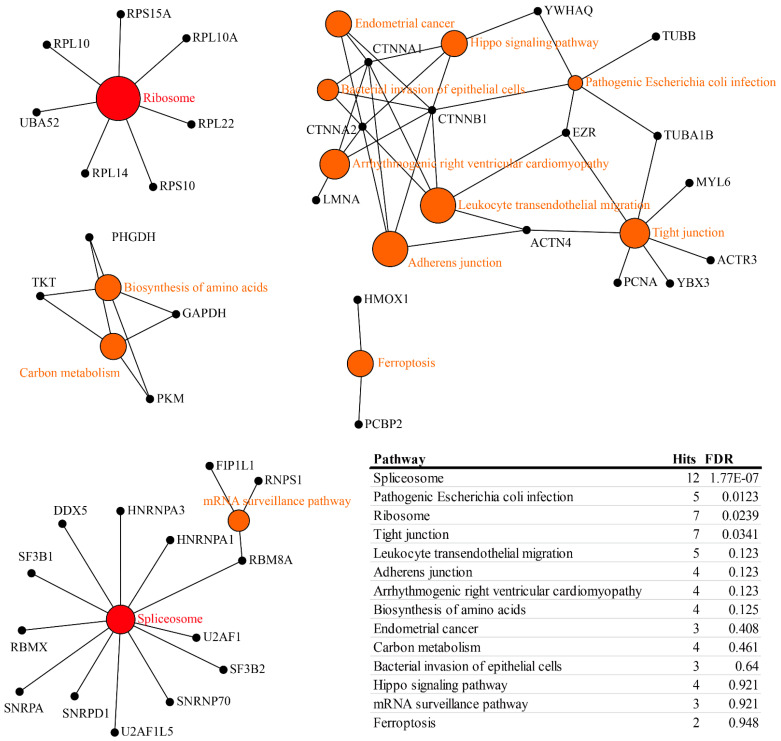
The enrichment network for differentially expressed proteins. Pathways were located in the center of networks, and the size of pathways were correlated with their enrichment score.

**Figure 6 molecules-27-01603-f006:**
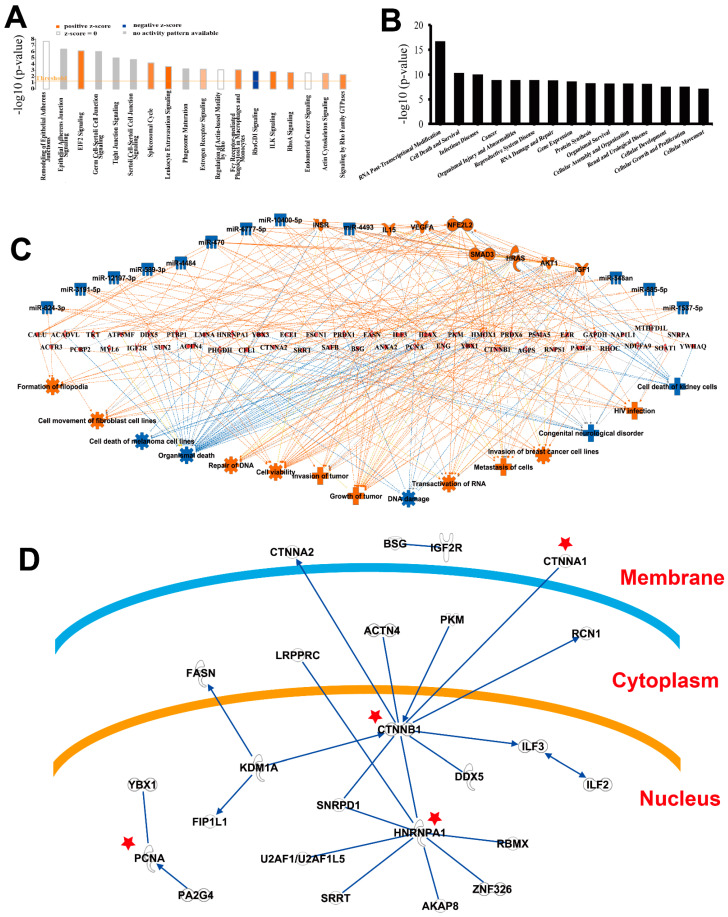
The IPA network analysis for differentially expressed proteins. (**A**) The enrichment pathway conducted by IPA. (**B**) The functions enriched by IPA analysis. RNA post-transcriptional modification is the most enriched function. (**C**) The most abundant three-layer molecular regulatory network in IPA. (**D**) The self-built network of differentially expressed proteins in IPA, that is, all the molecules in the network are differentially expressed proteins. Some proteins in the key positions of the network were found to be related to the tumor in IPA software and these are marked with asterisks.

**Table 1 molecules-27-01603-t001:** Key protein found in this study.

Symbol	Location	Type	Human Protein Atlas Biomarker Comments
PCNA	Nucleus	Enzyme	prognosis of oropharyngeal neoplasm
GAPDH	Nucleus	Enzyme	diagnosis of ovarian cancer
CTNNA1	Membrane	Other	diagnosis of gastric cancer.
CTNNB1	Nucleus	TR	diagnosis of Colorectal Cancer, thyroid cancer, gastric cancer, Cervical Cancer, mesothelioma
HNRNPA1	Nucleus	Other	
NOLC1	Nucleus	Transcription Regulator	Prognostic of thyroid cancer (unfavorable) and renal cancer (unfavorable)
SMHD1	Nucleus	Enzyme	Prognostic of renal cancer (unfavorable) and liver cancer (unfavorable)
NP1L1	Nucleus	Other	Prognostic of renal cancer (unfavorable) and liver cancer (unfavorable)
PHGDH	Cytoplasm	Enzyme	Prognostic of endometrial cancer (unfavorable) and glioma (favorable)

## Data Availability

The mass spectrometry proteomics data have been deposited to the ProteomeXchange Consortium (http://proteomecentral.proteomexchange.org, accessed on 17 February 2022) via the iProX partner repository with the dataset identifier PXD030920.

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
