# Peer review of "Comparative Proteomic Analysis of Drug Trichosanthin Addition to BeWo Cell Line"

_molecules, 2022, doi:10.3390/molecules27051603_

Round 1

Reviewer 1 Report

Yao et al report the effect of Trichosanthin treatment on BeWo cells, investigated using two different proteomic approaches. I believe that the overall design of the study is inappropriate, making the article not suitable for publication. This is due to the following major flaws:

  1. The authors used 2D gel-electrophoresis and MALDI-TOF-TOF-MS which is the traditional way of detection of differences in changes in protein level, though it is not quantitative. Then the authors describe a ‘label free method is an effective method that supplements 2D gel. Both methods are complementary to each other.’ Given that the method is not described or referenced it is difficult to judge if a label free quantification was actually carried out. To be able to quantify differences between two treatments the cells have to be grown with heavy or light isotopes of carbon in an amino acid, mixed with each other and then analysed using a mass-spec. instrument such as an LTQ Orbitrap described here. Such a procedure is not described here, indicating the proteomic analysis was likely not quantitative.
  2. The changes detected using mass-spec are not are not validated at protein or mRNA level. This would have resolved the lack of quantitative data from mass-spec analyses.
  3. While it would have been advantageous to use several cell lines for the proteomics, it may be too expensive. But the authors should have validated the differences detected using this method in other cell lines using other simpler methodology.
  4. The preparation of the drug, and validation of its purity is unclear, given that it is an extract. The MTT suggests poor dose response and 1 mg/ml is required to get a 50% reduction in cell viability. Also cell viability alone (MTT) is not a great measure of cell response to drug and should be combined with survival and apoptosis assays.
  5. Given the shortcomings in the design of the proteomic analyses, the detailed pathway analyses presented using various software are of less value.
  6. Bioinformatics analyses of proteomic data while useful, is only a guide for the possible mechanism of drug function. These data need to be validated using molecular biology techniques.
  7. The manuscript requires major revision with respect to language, experimental detail and structure.

Author Response

Yao et al report the effect of Trichosanthin treatment on BeWo cells, investigated using two different proteomic approaches. I believe that the overall design of the study is inappropriate, making the article not suitable for publication. This is due to the following major flaws:

  • The authors used 2D gel-electrophoresis and MALDI-TOF-TOF-MS which is the traditional way of detection of differences in changes in protein level, though it is not quantitative. Then the authors describe a ‘label free method is an effective method that supplements 2D gel. Both methods are complementary to each other.’ Given that the method is not described or referenced it is difficult to judge if a label free quantification was actually carried out. To be able to quantify differences between two treatments the cells have to be grown with heavy or light isotopes of carbon in an amino acid, mixed with each other and then analyzed using a mass-spec. instrument such as an LTQ Orbitrap described here. Such a procedure is not described here, indicating the proteomic analysis was likely not quantitative.

      The 2DE pathway focused on the image presentation of cellular proteome and only protein spots categorified as significantly changed were extracted for further analysis. Number of spots picked was limited by experimental design and practical considerations. The LC-MS/MS experiment was done undergoing shotgun method without heavy isotope labeling. The pathway could identify proteins as many as possible with few limitations. These experiments were complementary in content. The former experiment provided global view of protein expression pattern and underwent a quantification then identification methodology. The latter supplement an identification then quantitation way. These experiments were also complementary in methodology.

      The experimental section in the manuscript was inaccurate for the LC-MS/MS. We used in solution digestion and LC-MS/MS strictly following the MaxLFQ method described in 2014 by Cox et al. Briefly, equal micrograms of digested proteomes were analyzed and raw intensities automatically corrected by via optimization algorithm to generate LFQ values for proteins for further analysis. We corrected the experimental section and described the label free method used.

  • The changes detected using mass-spec are not validated at protein or mRNA level. This would have resolved the lack of quantitative data from mass-spec analyses.

      In the corrected manuscript, we made the right description of experiment and label free quantitation performed. We briefly summarized the correction in the previous answer. The MaxLFQ method was an accurate quantitation method. We followed the protocol and used strict criteria to ensure the data to be quantitative. TCS used in this research was prepared in one of the author’s lab and the procedure will be described in future publication. We could only acquire limited weight of powder. The experiments had consumed all the TCS and we cannot get material for further experiments.

  • While it would have been advantageous to use several cell lines for the proteomics, it may be too expensive. But the authors should have validated the differences detected using this method in other cell lines using other simpler methodology.

      The TCS we acquired was exhausted in the experiments and we made the description as detailed as possible. The MaxLFQ method was an accurate quantitation method and we used strict criteria to ensure the validity and accuracy. We put faith in the findings of the quantitative proteomic experiment on BEWO cells can withstand intense inspection. However, the reliability of this research did descend while we were not able to get enough material to validate the expression change in other cell lines.

  • The preparation of the drug, and validation of its purity is unclear, given that it is an extract. The MTT suggests poor dose response and 1 mg/ml is required to get a 50% reduction in cell viability. Also cell viability alone (MTT) is not a great measure of cell response to drug and should be combined with survival and apoptosis assays.

      TCS extraction procedure will be part of one of the authors’ future publication. The aim of MTT assay was to display the effectiveness of TCS in a distinctive way and the assay did, visually. This research was designed to find out the mechanism behind TCS killing cells. As explained in the previous answer, more assays’ design exceeded the materials we could possibly get. 

  • Given the shortcomings in the design of the proteomic analyses, the detailed pathway analyses presented using various software are of less value.

      We could only get limited amount of TCS. Given the portion used in the MTT assay, all the materials were put to as best use as possible. In general, the value of pathway analysis relies on the information provided. In this research, the pathway analysis not only validating the effect of TCS inactivating ribosome, but also indicating splicing and spliceosome participating in the TCS-cell interaction. This particular information is of great value.

  • Bioinformatics analyses of proteomic data while useful, is only a guide for the possible mechanism of drug function. These data need to be validated using molecular biology techniques.

      The bioinformatics analysis provided vital supplementary knowledge to the proteomics data. As the material we could possibly get was quite limited, those validations could only be the best potential target for further research.

  • The manuscript requires major revision with respect to language, experimental detail and structure.

      We modified the majority of the manuscript. The majority of changes were word choosing and sentence making. We corrected the mistakes and rewrote the ambiguous descriptions or misleading declarations. We are confident that big improvement has been made and the revised manuscript is ready for perusing.

Reviewer 2 Report

I recommend to homogenized all the text of the manuscript, the reference 21 do not support the cell culturing methodology. Quality of the figures must be increased.

Author Response

I recommend to homogenized all the text of the manuscript, the reference 21 do not support the cell culturing methodology. Quality of the figures must be increased.

We modified the majority of the manuscript. The majority of changes were word choosing and sentence making. We corrected the mistakes and rewrote the ambiguous descriptions or misleading declarations.

We followed the cell culture method from the website of ATCC, ref. 21 indirectly cited the website. We have changed ref. 21 to the website.

We generated SVG (Scalable Vector Graphics) format for each graph, which can be zoomed at any level, for this research and attached them in the manuscript .docx file. There could be some compression algorithm in the manuscript transfer system that caused the graph quality decay.

Reviewer 3 Report

This reviewer suggest extensive English edition to the authors. To understand the text is very hard. Text contains some pure scientific nonsenses.    

e.g. Line 90 - 92 – “…It can also rapidly deactivate the cell inner sugars, thus hindering the intracellular eggs. The synthesis of white leads to cell death, degeneration and necrosis.”

There are many vague statements throughout the whole manuscript.

e.g. Line 78 - Their molecular biological function can be analyzed and some signal pathways can be tried to be understood in this cancer.”

Line 80 - “The result provides the theoretical guidance for the study of treatment for some gyneco

logical diseases.”

In the introduction, information provided very often lack respective reference.

e.g. line 89 “In addition, the extract can act on placental syncytial trophoblasts.”

Experimental design and the description of used procedures is unclear and insufficient. Even after reading the Material method, section it not clear to this reviewer, what exact “label-free” methodology was actually used. Authors treated BeWo cells by the drug and measure the changes in protein expression patterns, however it is not clear to this reviewer, how does this supposed to lead us to conclusive effect on gynecological diseases?

e.g. Line 82 - “Besides BeWo cells can also be partially killed and can eventually be cured according to the result of MTT method.” …… What is the mechanism of partial killing?

The results overall are not properly described.

e.g. Figure 2. legend does not explain what is indicated in particular panels.

This reviewer could continue with the long list of things, which are not sufficiently addressed, described or concluded. In this form, this manuscript does not fulfil requirements for publication in any peer-reviewed journal. This reviewer suggest to extensively re-write the manuscript before considering the submission.  

Author Response

This reviewer suggests extensive English edition to the authors. To understand the text is very hard. Text contains some pure scientific nonsenses.

e.g. Line 90 - 92 – “…It can also rapidly deactivate the cell inner sugars, thus hindering the intracellular eggs. The synthesis of white leads to cell death, degeneration and necrosis.”

The misleading text was modified to properly describing TCS function.

There are many vague statements throughout the whole manuscript.

e.g. Line 78 - Their molecular biological function can be analyzed and some signal pathways can be tried to be understood in this cancer.”

Line 80 - “The result provides the theoretical guidance for the study of treatment for some gynecological diseases.”

The consecutive inaccurate text was modified to provide reasonable introduction to proteomics technology.

In the introduction, information provided very often lack respective reference. e.g. line 89 “In addition, the extract can act on placental syncytial trophoblasts.”

The missed reference was added.

Experimental design and the description of used procedures is unclear and insufficient. Even after reading the Material method, section it not clear to this reviewer, what exact “label-free” methodology was actually used. Authors treated BeWo cells by the drug and measure the changes in protein expression patterns, however it is not clear to this reviewer, how does this supposed to lead us to conclusive effect on gynecological diseases?

The experimental section in the manuscript was inaccurate for the LC-MS/MS. We used in solution digestion and LC-MS/MS strictly following the MaxLFQ method described in 2014 by Cox et al. Only BEWO cells were used as representative for trophoblast tumor and the phrase “gynecological diseases” were removed from in-proper sites.

e.g. Line 82 - “Besides BeWo cells can also be partially killed and can eventually be cured according to the result of MTT method.” …… What is the mechanism of partial killing?

The misleading text was modified. TCS can kill BEWO cells and this research provided a glance to the mechanism.

The results overall are not properly described. e.g. Figure 2. legend does not explain what is indicated in particular panels.

The ambiguous text was modified to explain the global presentation in figure 2.

This reviewer could continue with the long list of things, which are not sufficiently addressed, described or concluded. In this form, this manuscript does not fulfil requirements for publication in any peer-reviewed journal. This reviewer suggest to extensively re-write the manuscript before considering the submission.  

We modified the majority of the manuscript. The majority of changes were word choosing and sentence making. We corrected the mistakes and rewrote the ambiguous descriptions, vague statements or misleading declarations. We made improvement as much as we could. We are confident that the revised manuscript is ready for perusing.

Reviewer 4 Report

The authors performed a proteomics study for evaluation of trichosanthin (TCS) in the BeWo cell line. While the paper may have some interest to the scientific community, there are some major concerns to be addressed:

- The quality of the English is poor

- It is not possible to access to the raw data. The IPX0001296002 entry is related with other experiment. The authors should consider to submit the data to the PRIDE Proteomics database.

- The authors do not validate the found targets and do not present a justification for it.

- A justification for just two proteins have been found in common both in label free and 2DE has to be provided.

- It is necessary to perform a complementary gel free LC approach to improve the quality of the data.

Author Response

The authors performed a proteomics study for evaluation of trichosanthin (TCS) in the BeWo cell line. While the paper may have some interest to the scientific community, there are some major concerns to be addressed:

- The quality of the English is poor

We modified the majority of the manuscript. The majority of changes were word choosing and sentence making. We corrected the mistakes and rewrote the ambiguous descriptions, vague statements or misleading declarations. We made improvement as much as we could. We are confident that the revised manuscript is ready for perusing.

- It is not possible to access to the raw data. The IPX0001296002 entry is related with other experiment. The authors should consider to submit the data to the PRIDE Proteomics database.

We apologize for accidentally input the incorrect accession code for our raw data. The correct code is IPX0003967000.

- The authors do not validate the found targets and do not present a justification for it.

We followed the protocol of quantitative proteomics experiment and used strict criteria to ensure the data to be quantitative and the expression change significant and reliable. TCS used in this research was prepared in one of the author’s lab and the procedure will be described in future publication. We could only acquire limited weight of powder. The experiments had consumed all the TCS and we cannot get material for further experiments.

- A justification for just two proteins have been found in common both in label free and 2DE has to be provided.

The 2DE pathway focused on the image presentation of cellular proteome and only protein spots categorized as significantly changed were extracted for further analysis. Number of spots picked was limited by experimental design and practical considerations. The LC-MS/MS experiment was done undergoing shotgun method. The pathway could identify proteins as many as possible with few limitations. These experiments were complementary. The main reason was that only limited number of spots were cut for MALDI-TOF/TOF analysis.

- It is necessary to perform a complementary gel free LC approach to improve the quality of the data.

The experimental section in the manuscript was inaccurate for the LC-MS/MS. We used in solution digestion and LC-MS/MS strictly following the MaxLFQ method described in 2014 by Cox et al. Briefly, equal micrograms of digested proteomes were analyzed and raw intensities automatically corrected by via optimization algorithm to generate LFQ values for proteins. We corrected the experimental section and described the label free method used.

Round 2

Reviewer 1 Report

Yao et al have revised and met one of the concerns raised after the review of the article. The quantitative analyses of proteomics data are now clear.

Unfortunately, regarding the additional methodological clarification and mechanistical experimentation required, the authors have not satisfied the comments. While I sympathize the authors’ limitation in terms of access to reagents, I can only make a recommendation based on the scientific data presented.in the manuscript. The authors mention in their response ‘We put faith in the findings of the quantitative proteomic experiment on BEWO cells can withstand intense inspection.’ It is not reasonable that the readers can relay on this faith, and scientific evidence is required.

I hope that the authors find a way to revise the manuscript as per the comments made in the first round of revision, and provide experimental evidence to support their bioinformatics analyses.

Author Response

Yao et al have revised and met one of the concerns raised after the review of the article. The quantitative analyses of proteomics data are now clear.

Yes, we have revised the manuscript a lot, including English writing. Thanks for your affirmation.

Unfortunately, regarding the additional methodological clarification and mechanistical experimentation required, the authors have not satisfied the comments.

While I sympathize the authors' limitation in terms of access to reagents, I can only make a recommendation based on the scientific data presented.in the manuscript.

The authors mention in their response 'We put faith in the findings of the quantitative proteomic experiment on BEWO cells can withstand intense inspection.'

It is not reasonable that the readers can rely on this faith, and scientific evidence is required.

We could only get limited amount of TCS, which was product of extraction method development project. The MTT experiment was a requested condition. We used every proteomics and bioinformatics methods we were capable of to solidify the results. We used 2 proteomics methods to maximize the findings and each bioinformatic conclusion was manually checked. But for verification, there are no more TCS in our collaborator's lab to share. None of us is zealot, but unfortunately the data in manuscript were all the results we could possibly provide.

I hope that the authors find a way to revise the manuscript as per the comments made in the first round of revision, and provide experimental evidence to support their bioinformatics analyses.

Because of limited amount of TCS, we used two proteome method (2DE and LCMS) to focus the differential expressed proteins (DEPs). Volcano plot and OPLSDA were used to statistically filter the DEPs in respective method. And functional DEPs were screened by hub molecular in network with IPA Pathway and NetworkAnalyst. All the final DEPs have evidences related to cancer in literatures. Therefore, we think that we have adopted a reasonable way to screen the cancer-related proteins in TCS.

Reviewer 3 Report

The authors improved the quality of the English and thus increased the overall understandability of the manuscript. The matherial and methods section was rewritten and improved. The authors used label free MS method however any comparison or introduction into this topic is missing in the introduction. This reviewer suggests to briefly comment in the introduction section, on the recent paper describing novel label-free MS method published in Cells doi: 10.3390/cells10010068. After that this reviewer would consider publishing this manucript in the Molecules.

Author Response

The authors improved the quality of the English and thus increased the overall understandability of the manuscript. The material and methods section was rewritten and improved. 

Thanks for the affirmation, we have thoroughly rewrote the manuscript, including material and method part.

The authors used label free MS method however any comparison or introduction into this topic is missing in the introduction. This reviewer suggests to briefly comment in the introduction section, on the recent paper describing novel label-free MS method published in Cells doi: 10.3390/cells10010068. After that this reviewer would consider publishing this manuscript in the Molecules. 

In traditional label free method, not as label method (always use chemical, such as iTRAQ, SILAC and TMT,  to distinguish and quantify different samples), the high-resolution MS1 ion peaks were extracted as input for calculating the intensity-based quantitative features while the MS2 scans were often utilized merely to assign peptide IDs. Despite being used for more than a decade, there were booming breakthroughs in label-free method innovation (Cells doi: 10.3390/cells10010068), greatly improve the coverage of protein identification. However, the quantitative method of label free still uses the information of MS1 ion, and MaxQuant software implemented the label free quantitation calculation with MS1 ion.  So, like our method, there is no essential change with the "Cells" paper.

Reviewer 4 Report

It is still not possible to access the raw data with the provided accession code, IPX0003967000.   Without this information it is not possible to review this manuscript.

Author Response

It is still not possible to access the raw data with the provided accession code, IPX0003967000. Without this information it is not possible to review this manuscript.

We input the accession code in iProx repository and network turbulence hindered data acquisition. Fortunately, iProx is member of The ProteomeXchange Consortium and the dataset identifier is PXD030920.